# Changes in the Enteric Neurons Containing Selected Active Substances in the Porcine Descending Colon after the Administration of Bisphenol A (BPA)

**DOI:** 10.3390/ijerph192316187

**Published:** 2022-12-03

**Authors:** Krystyna Makowska, Sławomir Gonkowski

**Affiliations:** 1Department of Clinical Diagnostics, Faculty of Veterinary Medicine, University of Warmia and Mazury in Olsztyn, Oczapowskiego 14, 10-957 Olsztyn, Poland; 2Department of Clinical Physiology, Faculty of Veterinary Medicine, University of Warmia and Mazury in Olsztyn, Oczapowskiego 13, 10-957 Olsztyn, Poland

**Keywords:** enteric nervous system, porcine, bisphenol A, descending colon

## Abstract

Bisphenol A (BPA) is an endocrine disruptor widely distributed in the environment due to its common use in the plastics industry. It is known that it has a strong negative effect on human and animal organisms, but a lot of aspects of this impact are still unexplored. This includes the impact of BPA on the enteric nervous system (ENS) in the large intestine. Therefore, the aim of the study was to investigate the influence of various doses of BPA on the neurons located in the descending colon of the domestic pig, which due to similarities in the organization of intestinal innervation to the human gastrointestinal tract is a good animal model to study processes occurring in human ENS. During this study, the double immunofluorescence technique was used. The obtained results have shown that BPA clearly affects the neurochemical characterization of the enteric neurons located in the descending colon. The administration of BPA caused an increase in the number of enteric neurons containing substance P (SP) and vasoactive intestinal polypeptide (VIP) with a simultaneously decrease in the number of neurons positive for galanin (GAL) and vesicular acetylcholine transporter (VAChT used as a marker of cholinergic neurons). Changes were noted in all types of the enteric plexuses, i.e., the myenteric plexus, outer submucous plexus and inner submucous plexus. The intensity of changes depended on the dose of BPA and the type of enteric plexus studied. The results have shown that BPA may affect the descending colon through the changes in neurochemical characterization of the enteric neurons located in this segment of the gastrointestinal tract.

## 1. Introduction

Bisphenol A (BPA) is commonly used in the plastic industry. This substance is present in a great amount of everyday objects, such as bottles and containers, thermal paper, food can linings and others [1,2]. It is relatively well known that BPA may penetrate from plastics to the environment, food, and human and animal organisms, and adversely affects the functions of many internal systems [1,3,4].

The harmful effects of BPA is due to its similarity to estrogen and the ability to bind to estrogen receptors that are commonly found in various organs [5]. Therefore BPA-induced disorders are observed among others in reproductive organs, nervous and endocrine systems, liver, heart, and immune cells [1,2,6].

Due to the fact that the main route of exposure to BPA is the oral route, the gastrointestinal (GI) tract is significantly vulnerable for the harmful effects of this substance [1,7,8].

Although previous studies proved that BPA affects the GI tract [1,9,10,11,12], the knowledge about its effect on the enteric nervous system (ENS) within the large intestine is limited. On the other hand, it is known the ENS located in the wall of the GI tract is involved in the regulation of most functions of the intestines, including among others the adsorption of food components, motility, and secretory activity [13,14,15]. Therefore, the ENS, together with the intestinal immune system, makes a first barrier, which ingested toxicants have to defeat after entering the organism via the digestive system [9,10,11,12,13].

The organization of the ENS depends on the mammal species and the segment of the digestive tract [13,14]. In the descending colon of large mammals (such as the pig) it consists of three intramural plexuses: myenteric plexus (MP) in the muscle layer, between the longitudinal and circular muscular fibers, and the outer submucous plexus (OSP) located between the muscular layer and the mucosa and inner submucous plexus (ISP), which lays closer to the lumen of the gut, between the muscularis mucosa and lamina propria of the mucosal layer [15]. Until now, numerous neuronal active substances have been described in the enteric nervous structures. These substances may play roles of neurotransmitters, neuromodulators, intracellular enzymes, and/or transporting factors, and are involved not only in the regulation of intestinal functions in physiological conditions, but also participate in adaptive and/or protective reactions during pathological states, or under the impact of toxic substances [16,17,18,19,20,21].

One of these such substances, which may affect the ENS, is BPA. It is connected with the fact that estrogen receptors (to which BPA has affinity) are expressed not only on the enteric neurons, but also on the glial cells within the enteric ganglia and the interstitial cells of Cajal, which closely cooperate with the ENS in the regulation of intestinal motility [22,23]. Moreover, the influence of the activation of estrogen receptors on the functions and differentiation of enteric neurons have also been described [23,24]. These facts indicate that ENS may be vulnerable to the adverse impact of BPA, what has been confirmed in studies on the ENS in the small intestine [10,11,12].

However, there are no investigations on the influence of BPA on the ENS in the colon. On the other hand, it is known that the descending colon may be vulnerable to the impact of BPA contained both in the chyme and in peripheral blood supplying this part of the intestine, to a significant extent [25,26,27,28]. It is connected with mechanisms of the transition and metabolism of BPA in the body. Namely, after passing the small intestine, where BPA is subjected to the glucuronidation process, it is again deconjugated by microorganisms in the caecum and re-absorbed in the colon [26,27]. Moreover, the descending colon is the place in which various serious disease processes, including ulcerative colitis, colorectal cancer, or Crohn’s disease, may develop, and some previous studies have reported correlations between these diseases and the degree of exposure to BPA. In view of the above facts, knowledge concerning the impact of BPA on the colonic ENS may be the first step to better understand the pathological processes leading to colonic diseases.

Therefore, the aim of the present investigation was to describe for the first time the influence of various doses of BPA on the number of colonic enteric neurons containing selected neuronal factors, such as vesicular acetylcholine transporter (VAChT used as a marker of cholinergic neurons), galanin (GAL), vasoactive intestinal polypeptide (VIP), and substance P (SP), which are known as factors participating in the regulation of various intestinal functions [13,14].

Moreover, it is worth noting that the present study was performed on the domestic pig. The reason for choosing this animal species was due to the fact that porcine ENS is very similar to the human enteric nervous system, not only anatomically, but also neurochemically and electro-physiologically. Because of this fact, the domestic pig is an ideal animal model for studies on the influence of toxic substances on the human digestive tract [29], and the results obtained in this study may reflect changes taking place in the human ENS, after BPA administration.

## 2. Materials and Methods

Fifteen immature female pigs of Piétrain × Duroc race (8 weeks old, 20 kg body weight—b.w.) were used for the investigation. The animals were randomly divided into three equal groups of five pigs: control (C) group, low dose of BPA (LD) group, and high dose of BPA (HD) group. All pigs included into the study received gelatine capsules. Control animals received empty capsules, capsules given to animals from LD group contained BPA in a dose of 0.05 mg/kg b.w./day and capsules intended for pigs of HD group were filled with BPA in a dose of 0.5 mg/kg b.w./day.

During the study, animals were kept under standard laboratory conditions ensuring their welfare. The feed was given twice daily and drinking water was available ad libitum for the pigs of all groups. All activities performed during the investigation were conducted in agreement with the Local Ethics Committee in Olsztyn (Poland) (decision No 28/2013 of 22 May 2013 and No 65/2013/DLZ of 27 November 2013).

The mentioned above gelatine capsules were given once daily before morning feeding for 28 days. After this period, all animals were premedicated with Stresnil (Janssen, Beerse, Belgium, 75 μL/kg of b.w.) administered intramuscularly and then euthanised with an overdose of sodium thiopental (Thiopental, Sandoz, Kundl, Austria), given intravenously.

Immediately after the euthanasia about 2 cm long fragments of the descending colon (from the place where nerves from the inferior mesenteric ganglia supply the intestine) were collected from all animals and put into the 4% buffered paraformaldehyde (pH 7.4) for 1 h. For the next three days, the tissues were rinsed in a phosphate buffer and then stored in 18% phosphate-buffered sucrose at 5 °C for another 3 weeks. After this period the tissues were frozen at −22 °C and cut with a cryostat (HM 525, Microm International, Dreieich, Germany) to 14 µm thick fragments perpendicular to the lumen of the gut. The obtained intestinal slices were then attached to the microscopic slides and stored at −22 °C until further analysis.

Fragments of descending colon prepared in this way were subsequently subjected to the standard double immunofluorescence labelling described previously by Makowska and Gonkowski [20]. In brief, this method was performed as follows: after removing from freezer and 1 h drying in room temperature (rt), the colonic slices were treated with a “blocking” solution (10% normal goat serum, 0.1% bovine serum albumin, 0.01% NaN3, 0.25% Triton x-100 and 0.05% thimerosal in PBS) for another 1 h (rt). Then, a mixture of species specific primary antibodies was imposed on the tissue fragments. This step was performed in the humid chamber in rt and lasts overnight. In this study, a mixture consisted of two antibodies was used: the first of these was directed towards panneuronal marker—protein gene product 9.5 (PGP 9.5) and the second was directed towards one of the other neuronal active substance studied (i.e., SP, VIP, GAL or VAChT). Next day, the colonic fragments were incubated with a mixture of two species specific secondary antibodies (conjugated with fluorochromes Alexa Fluor 488 and Alexa Fluor 546) in the same conditions for 1 h. The list of all antibodies used in the study is presented in Table 1. At the end of labelling, the microscopic slides with colonic fragments were treated with buffered glycerol and covered with the coverslips. After every step of the immunofluorescent labelling method, the intestinal slices were rinsed in PBS (3 × 10 min). In order to validate the specificity of labelling the routine tests of antibodies including pre-absorption, omission, and a replacement test, were used.

The labelled colonic fragments were analysed using Olympus BX51 microscope equipped with the proper epi-fluorescence filter sets. The percentage of populations of neurons immunoreactive to particular substances (i.e., SP, VIP, GAL or VAChT) were evaluated in relation to neuronal cells positive for PGP 9.5 (used here as a marker of all neurons), number of which was considered as 100%. For this purpose, at least 500 cells containing PGP 9.5 in each type of the enteric plexus from each animal were evaluated for the presence of every other neuronal factor studied. Obtained results were pooled and presented as mean ± SEM. In order to obtain precise evaluation, only neurons with a clear visible nucleus were counted. Moreover, to prevent double counting of the same cells, colonic slices located at least 200 µm apart from each other were studied.

The statistical analysis of the obtained results was performed using a one-way analysis of variance (ANOVA) with Bonferroni’s Multiple Comparison post hoc test using a computer program Statistica 12 (StatSoft Inc., Tulsa, OK, USA). The results were considered statistically significant with *p* < 0.05 and very statistically significant with *p* < 0.01.

## 3. Results

During the present study, enteric neurons immunoreactive to all substances studied were found in every type of plexus in the descending colon of the control animals. The number of positive nerve cells clearly depended on the substance studied and the part of the enteric nervous system.

In the case of SP, the highest level of positive neurons reaching up to 22.60 ± 0.11% of all cells immunoreactive to pan neuronal marker PGP 9.5 was found in the ISP. In both OSP and MP, the number of such cells was lower, amounting to 17.33 ± 0.15% and 18.78 ± 0.15%, respectively (Figure 1). The population of VIP-LI enteric neurons was slightly smaller. In all three types of studied plexuses the number of VIP-positive nerve cells was similar and achieved 16.05 ± 0.1% in the ISP, 16.72 ± 0.18% in the OSP and 17.55 ± 0.22% in the MP (Figure 2). The number of neurons immunoreactive to GAL in the ISP amounted to 17.39 ± 0.14% of all PGP 9.5-positive cells and did not differ much from results revolving other neurotransmitters. However, in the other enteric plexuses the number of GAL-positive neurons was distinctly higher than the number of neuronal cells containing other neuronal factors and achieved 31.36 ± 0.13% in the OSP and even 34.30 ± 0.22% in the MP (Figure 3). In turn, the number of enteric neurons immunoreactive to VAChT (the marker of cholinergic nervous structures) in the ISP was highest amongst all of the neuronal populations studied and reached 26.11 ± 0.12% of all cells immunoreactive to PGP 9.5. In the OSP the percentage of VAChT-positive nerve cells was even higher amounting to 27.28 ± 0.16%, and in the MP the number of such neurons was slightly lower showing 24.28 ± 0.12% (Figure 4, Table 2).

During the present investigation, changes in the immunoreactivity of enteric neurons located in the porcine descending colon were found after the administration of both doses of BPA. These findings clearly depended on both the type of the enteric plexus, and the neuronal active substance studied. However, in every part of the ENS, and in the case of all neurotransmitters, the observed changes were statistically significant under the impact of both doses of BPA.

In the case of neurons immunoreactive to SP and/or VIP, the influence of high doses of BPA was more visible, and changes observed in the population of SP-LI neurons were less pronounced that those concerning neurons immunoreactive to VIP. The higher variations in the number of SP-positive neuronal cells were noted in the ISP, where the percentage of such neurons increased from 22.60 ± 0.11% in the control animals to 23.69 ± 0.12% and 30.27 ± 0.17% in the LD and HD groups, respectively (Figure 1). In turn, in the OSP low doses of BPA caused the increase the percentage of SP-LI neurons from 17.33 ± 0.15% in the healthy animals to 21.83 ± 0.10%, and high doses—to 24.65 ± 0.23%. The less visible changes in the number of SP-positive neurons were found in the MP, where the increase from 18.78 ± 0.15% in control animals to 20.88 ± 0.13% in the LD group and 24.21 ± 0.16% in the HD group was noted.

The fluctuations in the number of neurons immunoreactive to VIP were the most visible among all neuronal populations studied (Figure 2). In the ISP, the percentage of VIP positive neuronal cells increased from 17.55 ± 0.22% of all PGP 9.5; LI cells in control animals to 22.62 ± 0.17% and 29.82 ± 0.36% after the administration of low and high doses of BPA, respectively. Changes in the number of such neurons observed in the OSP where the most visible. In this type of enteric plexus low doses of BPA caused the increase in the number of VIP-LI cells from 16.72 ± 0.18% in control animals to 22.56 ± 0.22% and high doses of BPA, to even 31.96 ± 0.22%. In turn, after the exposure to BPA the number of VIP-LI neurons in the MP changed from 16.05 ± 0.1% to 22.67 ± 0.17% and 29.05 ± 0.17% in the LD and HD groups, respectively.

Moreover, the BPA administration also caused clear visible fluctuations in the population neurons immunoreactive to GAL and/or VAChT. However, contrary to the described above SP and VIP, the administration of BPA resulted in a decrease in the number of enteric neurons containing GAL and/or VAChT located in the wall of descending colon.

The more visible decrease under the impact of BPA was noted in the case of GAL-LI neuronal cells (Figure 3). Changes observed in the number of such neurons were more pronounced in the submucous plexuses. In the ISP the high doses of BPA caused the decrease in the percentage of GAL-LI neurons from 34.30 ± 0.22% of all PGP 9.5, LI cells to 22.25 ± 0.16%, and low doses of BPA, to 25.69 ± 0.14%. In turn, in the OSP the number of GAL-LI neurons decreased from 31.36 ± 0.13% to 27.15 ± 0.07% and 16.50 ± 0.19% under the impact of low and high doses of BPA, respectively. BPA also affected the population of neurons immunoreactive to GAL in the MP. In this plexus the number of such cells changed from 17.39 ± 0.14% in the control group to 14.72 ± 0.1% and 12.19 ± 0.07% in the LD and HD groups, respectively.

The abundance of population of VAChT-LI, neurons also changed after the administration of both doses of BPA (Figure 4). As in the case of GAL-positive neurons, the number of neuronal cells immunoreactive to VAChT also decreased under the impact of BPA. The most visible changes were noted in the OSP, where the number of VAChT—positive neurons decreased from 27.28 ± 0.16% in the control group to 22.28 ± 0.23% in animals received low doses of BPA and to 19.27 ± 0.11% in pigs treated with high doses of this substance. A decrease in the number of population of neurons immunoreactive to VAChT in the ISP was slightly lower. The percentage of such neurons in the ISP decreased from 24.28 ± 0.12% in control group to 20.33 ± 0.14% and 18.10 ± 0.13% in LD and HD groups, respectively. In turn, in the MP the low dose of BPA caused a decrease in the percentage of VAChT-positive neurons from 26.11 ± 0.12% in the control group to 23.33 ± 0.18%, and the high dose of BPA resulted in the decrease in the percentage of VAChT-LI neurons to 20.36 ± 0.15% of all PGP 9.5-positive cells. The summarization of results obtained in the present study is presented in the Table 2.

## 4. Discussion

First of all, the results obtained during the present investigation have shown that SP, VIP, GAL, and VAChT, are widely distributed in the ENS of the porcine descending colon. Such an observation is in agreement with earlier research [17,18,19]. The presence of neuronal factors studied in relatively high number of neurons (from above 16% to 34% of all neurons depending on the type of plexus and substance studied) located in all types of enteric plexuses may indicate the important roles of these substances in the regulation of various colonic functions.

In the light of previous studies, it is known that GAL plays multidirectional roles in the ENS, including the influence on secretion of other neurotransmitters from the enteric neurons, intestinal motility, and the secretory activity of the gastrointestinal mucosal layer [17,18,19,30]. In turn, a high level of neuronal cells immunoreactive to VAChT, which is the marker of cholinergic neurons, confirms a major role of acetylcholine—a key enteric neurotransmitter in the stimulation of intestinal motility and secretion [14,31]. Other substances studied in the present investigation, i.e., SP and VIP, are also involved in the regulation of various intestinal functions including intestinal motility, mucosal activity, immunological processes, blood flow, and sensory stimuli conduction [16,32,33,34,35,36,37,38].

The obtained results have also shown for the first time that BPA affects the neurochemical characterization of enteric neurons located in the wall of the porcine descending colon. Visible changes have been found in every type of the intramural plexuses after the administration of both doses of BPA, and their severity clearly depended on BPA dose and the type of the enteric plexus. Generally, these observations are in agreement with previous studies regarding the influence of toxic factors on the intestine, and confirm the ability of enteric neurons to change their neurochemical characterization under the influence of ingested toxicants [17,39,40].

The impact of BPA, which may bind to both type of estrogen receptors [41,42] on the colonic ENS observed in this study, results from the common occurrence of estrogen receptors on neurons and glial cells in the enteric plexuses (estrogen receptor β), as well as interstitial cells of Cajal taking part in the initiating of intestinal motility (Estrogen receptor α) [22,23].

As observed in the present study, BPA caused an increase in the number of neurons containing VIP and/or SP, and changes were the most visible in the case of VIP-positive neurons in the MP and ISP. Due to the multidirectional adverse effects of BPA and various functions of both these neuronal factors, it is rather difficult to determine the exact mechanism underlying the observed changes. Probably, it is connected with the functions of SP and VIP in adaptive and/or protective mechanisms, and other processes occurring under the toxic impact of BPA.

In the light of previous studies, SP, first of all, is a well-known active substance participating in sensory and pain conduction, which is widely distributed in both the central and peripheral nervous system, including the ENS [36,37,43]. Besides sensory activity, SP is actively involved in the regulation of intestinal motility and inflammatory processes, as well as in neuroprotective reactions [16,35,37,44]. The latter mentioned above function of SP is particularly visible in the central nervous system during neurodegenerative processes and traumatic brain injury [11,45,46]. However, it cannot be excluded that SP plays similar neuroprotective functions in the ENS, which is suggested by previous investigations [44,46]. In turn, VIP is an antioxidant and a major intestinal inhibitory factor affecting gastrointestinal motility and secretion [13,15,16]. Moreover, VIP also has anti-inflammatory and neuroprotective properties [33,34,38,47,48]. Interestingly, in some cases, the functions of VIP and SP are opposed to each other (for example, influences on the motility or immune cells) [14,16,49,50], and it is not clear why changes in the number of enteric neurons containing SP and VIP induced by BPA are similar, and consist in the increase in the number of both neuronal populations. However, similar situations were noted during the influence of other pathological and/or toxic factors on the ENS.

During the present study, it has been found that BPA caused a decrease in the number of neurons containing VAChT and/or GAL, in all types of the colonic enteric plexuses.

The decrease in the number of VAChT-LI neuronal cells has been also noted in previous studies on the influence of BPA on the ENS in the small intestine [10,11,12,39]. The decrease in the number of neurons containing VAChT, on the one hand, may indicate that cholinergic neuronal cells, which are crucial in regulating bowel functions in physiological states, are less important under the impact of pathological factors. Therefore, the inhibition of acetylcholine synthesis appears to aid the synthesis of other neuronal substances playing more important roles in adaptive and/or protective reactions. On the other hand, alterations in the number of VAChT-LI neurons may be caused by dysfunctions in colonic motility and secretory activity provoked by BPA administration.

In the case of GAL, the situation is more complicated. In the light of previous studies, the functions of GAL clearly depend of the segment of the GI tract. Next to the neuroprotective properties, especially visible during ischemia and neuro-degenerative processes, but also noted in gastrointestinal diseases [51,52], GAL regulates intestinal secretion, motility, blood flow, and immunological processes, and affects intestinal absorption [17,18,19,53]. Therefore, the observed variations in the number of GAL-LI neurons may also be an effect of BPA-induced disorders in the functionality of the intestinal barrier. In turn, acetylcholine is the main neurotransmitter in the ENS under physiological conditions showing stimulatory activity [14,31]. The decrease in the number of GAL positive neurons noted in the present study is not in agreement with previous studies, in which BPA caused an increase in the number of neuronal cells containing GAL in the wall of the small intestine [10,12]. The reason for this difference is not clear, but on the one hand it may be connected with differences in GAL functions in various segments of the GI tract known from previous investigations [31,54], and on the other hand from undefined differences in the BPA impact between the small and large intestine.

It should be pointed out that the BPA influence on the GI tract is multidirectional. It has been reported that BPA intoxication changes the intestinal microbiome [9] and motility [49]. The subsequent BPA-induced decrease in the GI tract motility may be one of the mechanisms facilitating absorption of this substance, resulting in increased systemic toxicity [55]. It is also known that the majority of colonic diseases, such as ulcerative colitis, cancer, or aganglionosis, are accompanied by disturbances in colonic motility, and these disturbances may result from exposure to BPA [49,56,57]. This has been confirmed by previous studies, which have reported the correlation between the severity of exposure to BPA and some pathological processes in the descending colon [58]. It should be underlined that BPA not only affects the intestinal motility, but also increases mucosal permeability through fracturing the intestinal barrier [9], as well as inducing apoptosis and mitochondrial dysfunction in the intestinal mucosal layer, which is connected with BPA-induced oxidative stress and inflammatory processes [9,56]. All of the above mentioned effects of BPA action may reflect in changes in the neurochemical characterization of the enteric neurons observed in the present study.

Moreover, these changes may also result from inflammation and apoptosis processes, as well as oxidative stress caused by BPA [56]. Admittedly, during the present investigation, the inflammatory changes after the administration of BPA has not been observed in the descending colon, but it cannot be excluded that changes in neurochemical characterization result from the pro-inflammatory properties of BPA, and they are the first signs of subclinical inflammation. The observed changes may also be connected with sensory and pain stimuli conduction, but it is less likely because the doses of BPA used in the present study were relatively low, and the experimental animals did not show any symptoms of pain. However, due to the limitation of the immunofluorescence method, the exact explanation of direct reasons and mechanisms of change in the number of enteric neurons containing various neuronal factors observed in the present study, is practically impossible at this stage and requires further research on this issue.

Nevertheless, the obtained results clearly indicate that BPA affects the ENS not only in the small intestine, which is known from previous studies [10,11,12], but also in the descending colon. This fact may be closely connected with the metabolism and disposition of BPA entering the organism via the GI tract [59], in which participation of the colon is particularly important. The majority of BPA contained in food is subjected to the glucuronidation process in the enterocytes of the small intestine, with the participation of UDP-glucuronosyltransferase enzymes [26]. The BPA-glucuronide (BPA-GA) formed in this process shows definitely smaller estrogenic activity; therefore, glucuronidation is considered to be a form of the body’s defense against the adverse effects of BPA [26,27,55]. Most of the BPA-GA synthetized in the enterocytes is transported back into the intestinal lumen. The rest of this metabolite, and the remaining free BPA, are absorbed into the blood and transported into the liver. It should be pointed out that the amount of BPA, which is not transformed into BPA-GA in the enterocytes, dramatically increases when the doses of BPA given orally are high. The glucuronidation process of free BPA is also carried out in the liver, and BPA-GA is secreted to the bile and transported to the GI tract. BPA-GA excreted to the lumen of the intestine flows together with intestinal content and may be subjected to renewed deconjugation to free BPA by gut microbiota, located mainly in the caecum. Therefore, the descending colon is extensively vulnerable to the impact of BPA. This fact is confirmed by previous studies, which have described a high level of free BPA in the lumen of colon [55], as well as by the results obtained in the present study, in which the clear influence of BPA administration on the ENS has been noted. Moreover, in the light of latest investigations, it is also known that the glucuronited metabolite of BPA synthetized as mentioned above, as a form of the body’s defense against the endocrine disrupting effects of BPA in the enterocytes and liver, and then excreted to the intestine lumen both by enterocytes as well as with bile, may show some negative influence on the organisms. Namely, BPA-glucuronide affects, among others, the glycolysis and functional responses of neutrophils and may show even higher endocrine-disrupting potential than BPA [60]. So, it cannot be excluded that the changes noted in this study are connected not only with the impact of free BPA, but also its glucuronited metabolite.

During the present investigation, changes in the neurochemical characterization of the colonic enteric neurons were observed under the impact of both doses of BPA used in the study. It should also be highlighted that the dose of 0.05 mg/kg body weight/day for a long time was considered by the European Food Safety Authority (EFSA) as a safe tolerable daily intake (TDI) of BPA [28], and until the present day, such a dose is recommended as a TDI, or reference dose, in some countries [61]. Changes in the colonic ENS noted in the present study clearly show that even such a low dose of BPA, which does not cause any symptoms of intoxication, affects the neurochemical characterization of the enteric neurons. These observations, together with previous studies, in which the influence of similar doses of BPA on the autonomic nervous system supplying various internal organs has been reported, strongly suggest that the decision of EFSA [28] on the reduction of TDI for BPA from 0.05 mg to 4 µg/kg b.w./day, was justified. Comparing the doses used in this study with the average daily human exposure to BPA in “everyday life”, it should be stated that doses used in this investigation are higher. However, daily human exposure to BPA depends on numerous factors, such as diet, profession, the place of residence, and the degree of its pollution, and in some situations (for example in the case of numerous old-style dental fillings that contain BPA, or in the case of some professions) human exposure may be even higher than doses used in this study [1,2,3,4].

## 5. Conclusions

To sum up, BPA given orally causes changes in the number of enteric neurons containing VAChT, GAL, VIP, and SP, located in the ENS of the porcine descending colon. This observation strongly suggest that BPA affects not only the innervation of the anterior segments of the GI tract, where BPA is mainly metabolized, but also the large intestine. Moreover, obtained results clearly show that even relatively small doses of BPA, that do not cause symptoms of poisoning, may affect the colonic enteric neurons located in the descending colon. Probably, the observed changes result from the neurotoxic and/or pro-inflammatory activity of BPA, and are connected with adaptive and/or protective reactions. However, due to the limitations of the immunofluorescence technique, the explanation of the exact causes and mechanisms underlying the observed changes is practically impossible, and remains for further comprehensive studies.

## Figures and Tables

**Figure 1 ijerph-19-16187-f001:**
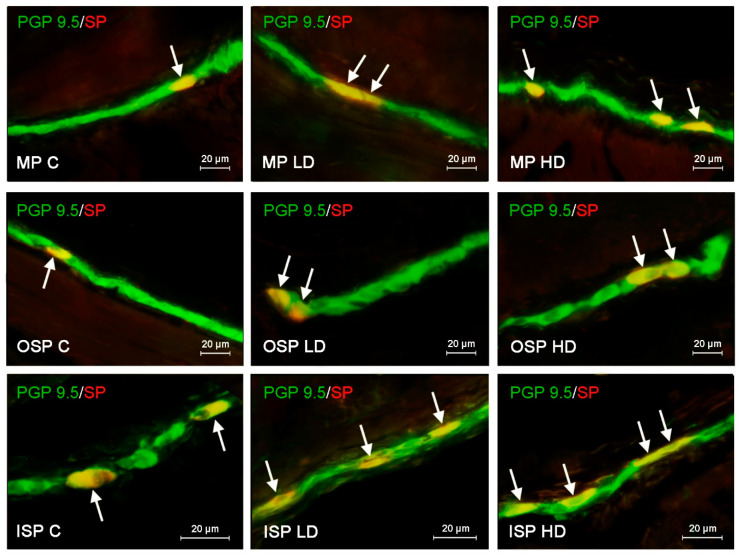
Distribution pattern of nerve cells immunoreactive to protein gene-product 9.5 (PGP 9.5)–used as pan neuronal marker and substance P (SP) in the myenteric plexus (MP), outer submucous plexus (OSP) and inner submucous plexus (ISP) of porcine descending colon under physiological conditions (C) and after administration of small dose (LD) and high dose (HD) of bisphenol A. The pictures are the result of the overlap of both stainings. The arrows point neurons immunoreactive for both–PGP 9.5 and SP.

**Figure 2 ijerph-19-16187-f002:**
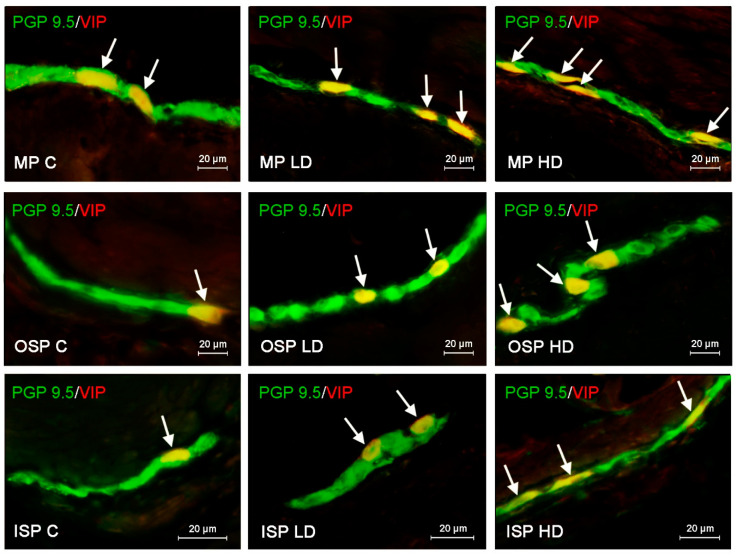
Distribution pattern of nerve cells immunoreactive to protein gene-product 9.5 (PGP 9.5)–used as pan neuronal marker and vasoactive intestinal polypeptide (VIP) in the myenteric plexus (MP), outer submucous plexus (OSP) and inner submucous plexus (ISP) of porcine descending colon under physiological conditions (C) and after administration of small dose (LD) and high dose (HD) of bisphenol A. The pictures are the result of the overlap of both stainings. The arrows point neurons immunoreactive for both–PGP 9.5 and VIP.

**Figure 3 ijerph-19-16187-f003:**
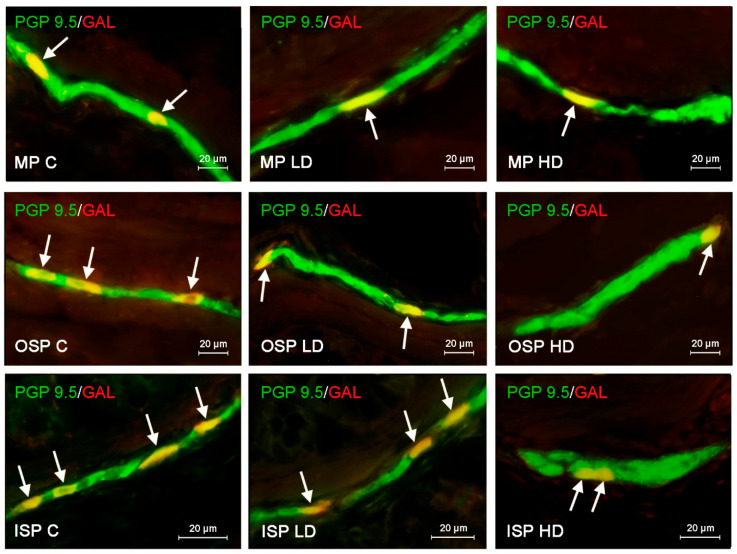
Distribution pattern of nerve cells immunoreactive to protein gene-product 9.5 (PGP 9.5)–used as pan neuronal marker and galanin (GAL) in the myenteric plexus (MP), outer submucous plexus (OSP) and inner submucous plexus (ISP) of porcine descending colon under physiological conditions (C) and after administration of small dose (LD) and high dose (HD) of bisphenol A. The pictures are the result of the overlap of both stainings. The arrows point neurons immunoreactive for both–PGP 9.5 and GAL.

**Figure 4 ijerph-19-16187-f004:**
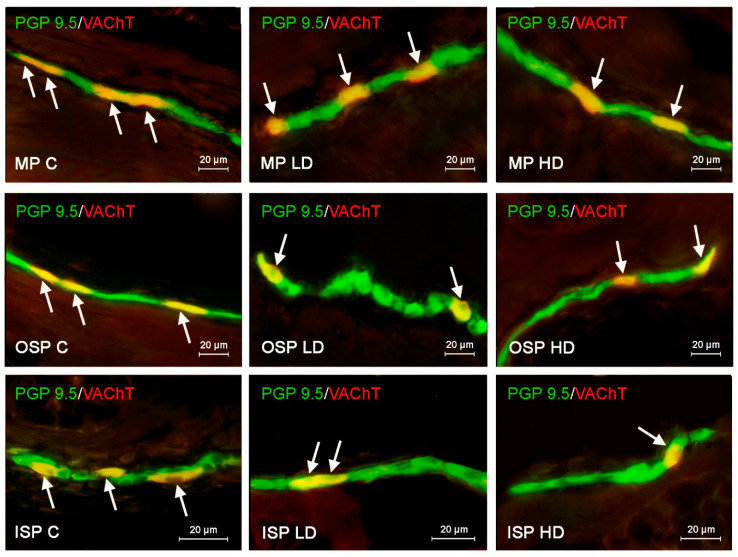
Distribution pattern of nerve cells immunoreactive to protein gene-product 9.5 (PGP 9.5)–used as pan neuronal marker and vesicular acetylocholine transporter (VAChT) in the myenteric plexus (MP), outer submucous plexus (OSP) and inner submucous plexus (ISP) of porcine descending colon under physiological conditions (C) and after administration of small dose (LD) and high dose (HD) of bisphenol A. The pictures are the result of the overlap of both stainings. The arrows point neurons immunoreactive for both–PGP 9.5 and VAChT.

**Table 1 ijerph-19-16187-t001:** List of antisera and reagents used in the immunohistochemical investigations.

Primary Antibodies				
Antigen	Code	Species	Working dilution	Supplier
PGP 9,5	7863-2004	Mouse	1:1000	Biogenesis Ltd., Poole, UK
SP	8450-0505	Rat	1:1000	Bio-Rad (AbD Serotec), Kidlington, UK
VIP	VA 1285	Rabbit	1:2000	Enzo Life Sciences; Farmingdale, NY, USA
GAL	T-5036	Guinea Pig	1:2000	Peninsula
VAChT	H-V006	Rabbit	1:2000	Phoenix Pharmaceuticals
**Secondary Antibodies**				
Reagents			Working dilution	Supplier
Alexa fluor 488 donkey anti-mouse IgG			1:1000	Invitrogen, Carlsbad, CA, USA
Alexa fluor 546 donkey anti-rabbit IgG			1:1000	Invitrogen
Alexa fluor 546 donkey anti-rat IgG			1:1000	Invitrogen
Alexa fluor 546 donkey anti-guinea pig IgG			1:1000	Invitrogen

**Table 2 ijerph-19-16187-t002:** The distribution of enteric neurons immunoreactive to substance P (SP), vasoactive intestinal polypeptide (VIP), galanin (GAL) and vesicular acetylcholine transporter (VAChT) in the porcine descending colon.

		C	LD	HD
SP	MP	18.78 ± 0.15 *	20.88 ± 0.13 *	24.21 ± 0.16 *
	OSP	17.33 ± 0.15 *	21.83 ± 0.1 *	24.65 ± 0.23 *
	ISP	22.60 ± 0.11 *	23.69 ± 0.12 *	30.27 ± 0.17 *
VIP	MP	16.05 ± 0.1	22.67 ± 0.17 *	29.05 ± 0.17 *
	OSP	16.72 ± 0.18	22.56 ± 0.22 *	31.96 ± 0.22 *
	ISP	17.55 ± 0.22	22.62 ± 0.17 *	29.82 ± 0.36 *
GAL	MP	17.39 ± 0.14	14.72 ± 0.1 *	12.19 ± 0.07 *
	OSP	31.36 ± 0.13	27.15 ± 0.07 *	16.50 ± 0.19 *
	ISP	34.30 ± 0.22	25.69 ± 0.14 *	22.25 ± 0.16 *
VAChT	MP	26.11 ± 0.12	23.33 ± 0.18 *	20.36 ± 0.15 *
	OSP	27.28 ± 0.16	22.28 ± 0.23 *	19.27 ± 0.11 *
	ISP	24.28 ± 0.12	20.33 ± 0.14 *	18.10 ± 0.13 *

Values are presented as the relative frequency of SP-, VIP-, GAL- or VAChT- immunoreactive neurons presented as % (mean ± SEM) in relation to all counted nerve cell bodies in the myenteric plexus (MP), outer submucous plexus (OSP) and inner submucous plexus (ISP). Statistically significant differences (*p* < 0.05) between control animals (C group) and animals receiving low dose of BPA (LD group) as well as between C group and animals treated with a high dose of BPA (HD group) are marked with *.

## Data Availability

Data is contained within the article.

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
