# Peer review of "Changes in the Enteric Neurons Containing Selected Active Substances in the Porcine Descending Colon after the Administration of Bisphenol A (BPA)"

_ijerph, 2022, doi:10.3390/ijerph192316187_

Round 1
Reviewer 1 Report
Interesting, sound study which is also well presented.
Before recommending for publication, the Authors should clarify some points
1: To my best knowledge, the colon is especially rich in ER-beta receptors. The Authors should consider whether the interactions of BPA with different ER nuclear receptors influence the observed effects
2: the Authors make reasonable considerations on the TDIs for BPA in relation to the dose levels selected in their study (lines 393-404). Meanwhile the Authors should also consider the study dose levels in comparison with the actual oral intakes of BPA in different world areas. Are the dose levels relevant to human exposure levels?
3. While the reasoning presented on BPA-glucuronide is sound (line 372-392), are we entirely sure that glucuronidation of BPA is a a protective mechanism in 100% of cases? For instance see https://pubmed.ncbi.nlm.nih.gov/33091430/ . Some additional considerations about this issue?
Author Response
The authors thank the Reviewer for evaluation of the manuscript.
Interesting, sound study which is also well presented.
Before recommending for publication, the Authors should clarify some points
1: To my best knowledge, the colon is especially rich in ER-beta receptors. The Authors should consider whether the interactions of BPA with different ER nuclear receptors influence the observed effects
Information about estrogen receptors in the ENS has been added in the introduction and discussion according to suggestions of the Reviewer, lines 64-71 and 297-301.
2: the Authors make reasonable considerations on the TDIs for BPA in relation to the dose levels selected in their study (lines 393-404). Meanwhile the Authors should also consider the study dose levels in comparison with the actual oral intakes of BPA in different world areas. Are the dose levels relevant to human exposure levels?
Information about exposure of humans to BPA in “everyday” live has been added at the end of discussion, lines 418-424.
3. While the reasoning presented on BPA-glucuronide is sound (line 372-392), are we entirely sure that glucuronidation of BPA is a a protective mechanism in 100% of cases? For instance see https://pubmed.ncbi.nlm.nih.gov/33091430/ . Some additional considerations about this issue?
The authors are in agreement that BPA-glucuronide is not safe to organisms and (like BPA) has toxic properties. Information about this issue has been added in discussion lines 398-406.

Reviewer 2 Report
The authors investigated the effects of BPA exposure on active substances in enteric nervous neurons. This is an interesting topic, but there are some issues that should be revised in this paper.
1. The abstract should be an description to the topic and problems to be solved, not a simple repetition of the introduction. It is suggested that the abstract be rewritten to highlight the topic of the article.
2. The introduction is confusing. For example, line 36 mentions that BPA is an environmental estrogen, and BPA causes widespread harm to organisms due to its ubiquitous receptors. But how does it relate to the enteric nerves? Also, lines 70 -72 are repeated with lines 42- 44.
3.Lines 95-96, what does “indented neutral for the living organisms” mean?
4. The results part of this study is less, just a simple description of the table 2, which is not quite consistent with the discussion part. In the discussion part, the author should discuss the results more deeply.
5. Variations in neuronal neurochemistry are mentioned in lines 322-323, and this is closely related to the active substance in this study. According to the results in the study, BPA caused changes in these active substances, but did not lead to variations in neuronal neurochemistry. How to confirm the link between the two kinds of changes?
6. The structure of this article is chaotic. Should table 1 be placed above the results section? We recommend consulting the journal's author's guide to determine the correct placement of the table.
7. The language of the article should be polished.
8. Whether the double immunofluorescence technique alone can prove the damage of BPA to the enteric nervous neurons containing selected active substances? Additional experiments should be supplement to further prove the conclusion.
Author Response
The authors thank the reviewer very much for review.
The authors investigated the effects of BPA exposure on active substances in enteric nervous neurons. This is an interesting topic, but there are some issues that should be revised in this paper.
- The abstract should be an description to the topic and problems to be solved, not a simple repetition of the introduction. It is suggested that the abstract be rewritten to highlight the topic of the article.
According to the Reviewer’s suggestion, the abstract has been significantly rewritten.
- The introduction is confusing. For example, line 36 mentions that BPA is an environmental estrogen, and BPA causes widespread harm to organisms due to its ubiquitous receptors. But how does it relate to the enteric nerves? Also, lines 70 -72 are repeated with lines 42- 44.
The authors are in agreement with the Reviewer that old version of the introduction was a bit chaotic. This was because the authors wanted to convey as much information as possible. The authors thank for drawing attention to this issue. The introduction has been significantly rewritten. All suggestions of the Reviewer have been into account. The authors hope that now the introduction is more readable
3.Lines 95-96, what does “indented neutral for the living organisms” mean?
This sentence has been removed during complete rewording of the introduction
- The results part of this study is less, just a simple description of the table 2, which is not quite consistent with the discussion part. In the discussion part, the author should discuss the results more deeply.
The results have been written typically for this type of manuscripts. The table has been added as a summarization of the results so that the reader can quickly familiarize himself with the main results of the experiment. In turn discussion part has been completely rewritten to be consistent with results. According to suggestion of the Reviewer results in new version are discussed more deeply.
- Variations in neuronal neurochemistry are mentioned in lines 322-323, and this is closely related to the active substance in this study. According to the results in the study, BPA caused changes in these active substances, but did not lead to variations in neuronal neurochemistry. How to confirm the link between the two kinds of changes?
The authors are in agreement with the Reviewer that this fragment of old version of the manuscript was not clear. During the reedition of discussion this fragment has been removed. The immunofluorescence technique used in this study allows to observe the presence of particular active substances in neurons and changes in the number of these neurons. The changes in number of particular neuronal populations may suggest that neurochemical characterization of neurons (neurochemical profile) undergoes changes.
- The structure of this article is chaotic. Should table 1 be placed above the results section? We recommend consulting the journal's author's guide to determine the correct placement of the table.
As mentioned above introduction and discussion have been significantly rewritten. The authors hope that now the manuscript is less chaotic. On the other hand Table 1 represents the specifications of antibodies used in the double immunofluorescence technique so it belongs to the Materials and Method section. It is in agreement with the authors guideline of the journal specifying that Materials and Methods section should be above the Results section. The authors decided to include such a table instead of listing the individual parameters of all the antibodies used in the text to make the manuscript more readable.
- The language of the article should be polished.
According to the Reviewer’s suggestion, some moderate language changes have been made in the manuscript.
- Whether the double immunofluorescence technique alone can prove the damage of BPA to the enteric nervous neurons containing selected active substances? Additional experiments should be supplement to further prove the conclusion.
Of course, the double immunofluorescence technique alone is not enough to prove BPA damage to nerve neurons, because we know that changes in the expression of active substances in those neurons is related to their neuroplasticity in response to the influence of the studied substance but we do not know the exact mechanism of these changes. Such neurochemical changes may be related not only to the mechanical damage of cells, therefore further research is required to investigate the exact mechanism of BPA's action. The authors highlighted these limitations of their work in the discussion section. The authors are in agreement with the Reviewer that further studied are needed to exact explanation of all aspects connected with the BPA impact on the colonic ENS (this fact is also underlined in discussion). On the other hand the manuscript is the first study on BPA-induced changes in the innervation of the colon and may be a first step to further investigations and projects in the future. Unfortunately the project is ended and the inclusion of new methods would entail the need to repeat the entire experiment, including euthanasia of new animals, which is practically impossible for ethical reasons of animal protection, Simultaneously the conclusion has been rewritten and now it reflects the obtained results.
The authors hope that corrections and clarifications will allow the manuscript to be published.
